# A Change of Hallmark: An Update

**DOI:** 10.3390/cells14191490

**Published:** 2025-09-24

**Authors:** Tom Donnem, David Kerr, Leonid L Nikitenko, Francesco Pezzella

**Affiliations:** 1Department of Oncology, University Hospital of North Norway, N-9038 Tromso, Norway; tom.donnem@uit.no; 2Institute of Clinical Medicine, UiT The Arctic University of Norway, N-9037 Tromso, Norway; 3Nuffield Divisio Medical Labaratory Science, Radcliffe Department of Medicine John Radcliffe Hospital, Oxford OX3 9DU, UK; david.kerr@ndcls.ox.ac.uk; 4Centre for Biomedicine, Hull York Medical School, University of Hull, Hull HU6 7RX, UK; l.nikitenko@hull.ac.uk

**Keywords:** co-option, angiogenic, non-angiogenic, tumours, primary, metastases

## Abstract

We review the latest development in non-angiogenic tumours. We focused on the last 3 years except the rarer tumours, for which the papers are older. Following the explanation of the modified hallmark of cancer, inducing angiogenesis and/or accessing vessels, the authors review primary and metastatic tumours growing into lung, liver and brain, plus oral cancer, lymphomas and node metastasis. Also progress in treatment, not many unfortunately, and techniques in non-angiogenic tumours are discussed.

## 1. Introduction

In 2011, Hanahan and Weinberg considered “inducing angiogenesis” a hallmark of cancer [1]. Eleven years later, in 2022, Hanahan described a slightly revised hallmark: “inducing or accessing vasculature” [1]. This change may not seem to mark the end of an era, but the term “accessing” signals the end of the idea that any cancer can be treated with drugs aimed solely at preventing new vessels from sprouting [2]. In a previous review [3], we showed how the exploitation—or “co-option”—of pre-existing normal vessels has evolved and impacted oncology, as the old paradigm of a single anti-angiogenic drug that is effective across all tumours no longer holds (Figure 1). Furthermore, it should not be forgotten that several angiogenic tumours begin with the co-option of pre-existing vessels rather than as avascular masses. This concept was introduced by Holash and colleagues in 1999 [4]. These co-opted vessels subsequently regress, the neoplastic mass becomes avascular, and only then is neo-angiogenesis triggered. Here, we present an update, focusing primarily—but not exclusively—on developments from the past three years.

## 2. Co-Option in the Lung: Primary Tumours and Metastases

Non-angiogenic tumours in the lung were first described in the 1990s, including both primary and metastatic lung carcinomas [5], and were discovered by chance. The story goes that a thoracic surgeon with a strong interest in lung tumours aimed to review all known prognostic factors published for non-small-cell lung carcinoma [6]. At the time, angiogenesis—measured as micro-vessel density—was considered one of the most important prognostic factors and was, therefore, proposed for inclusion in the study. The surgeon agreed on the condition that the scoring be performed by the proponent, a pathologist. Micro-vessel density turned out to be negative in the studied series, as were most other prognostic factors. However, during the scoring process, vascular co-option was identified. Only the tumours lacking angiogenesis had a worse prognosis in the lung [6,7,8]. This unfavourable prognosis has since been observed across many tumours identified as non-angiogenic.

A recent study of lung cancer [9] examined different histological subtypes and demonstrated that in primary non-small-cell lung carcinomas—specifically adenocarcinomas but not squamous-cell carcinomas—tumours with more than 5% co-option exhibit the worst prognosis. In this context, co-option serves as an independent negative prognostic factor. The authors found that 18% of cases displayed predominant co-option, while 40.1% had more than 5% co-option; the latter group also had significantly worse outcomes. Co-opting tumours were positively correlated with platelet-derived growth factor receptor beta (PDGFRβ), a receptor tyrosine kinase that, among other functions, promotes angiogenesis. Its expression was observed in the stroma; however, no association with angiogenic markers was found in either angiogenic or non-angiogenic tumours.

Co-option did not correlate with glucose transporter type 1 (GLUT1) expression. Immunomarkers such as CD3, CD4, CD8, CD45RO, PD1, and CD20 had prognostic value in adenocarcinomas with less than 5% co-option and were also prognostic in squamous-cell tumours [9].

A 3D reconstruction of lung tissue with adenocarcinoma was also performed [10]. While the primary aim was to study the spread through air spaces (STASs), the authors observed that, once in a new alveolar space, tumour cells survive by adhering to pre-existing vessels. They concluded that the tumour cells survive via co-option. This study further supports the findings of an earlier study [11], which was initially conducted to acquire further evidence of co-option in primary lung carcinomas.

Metastases of the lung have also been studied. In 2025, Torre-Cea et al. published a review describing how co-opted vessels, metastatic cells, and basement membranes interact within the lung microenvironment. Integrin-mediated mechanisms appeared to be particularly important for co-option. Subsequent hypoxia triggered the recruitment of cancer-associated fibroblasts (CAFs), the production of transforming growth factor beta (TGF-β), a cytokine also involved in migration, and increased extracellular matrix rigidity—changes that accelerate metastatic spread [12]. In contrast, angiogenesis was promoted in the presence of collagen type IV, which supports endothelial cell migration and proliferation, ultimately resulting in new intra-tumoral, but co-opted, vessels [12].

Teuwen et al. [13] further studied lung metastases using a mouse model, injecting RENCA (RENal CAncer) cells into the tail veins of mice with or without Sunitinib on days 10, 21, and 36. By day 21—after 10 days of Sunitinib—70% of metastases were non-angiogenic, while only 30% were angiogenic (compared to 70% angiogenic and 30% non-angiogenic in controls). Long-term treatment (36 days) with Sunitinib resulted in a predominance (75%) of non-angiogenic metastases. Although untreated mice had to be euthanized earlier due to tumour burden, treatment with anti-angiogenic agents slightly improved overall survival but not tumour-free survival, as the non-angiogenic tumour burden rapidly increased (comprising 75% of growing metastases). Tumour endothelium single-cell sequencing revealed that co-opted endothelial cells lacked proliferation and tip cell markers. Similarly, pericytes were quiescent. The authors found it surprising that neither endothelial cells nor pericytes showed activation—whether in spontaneous non-angiogenic tumours or after Sunitinib treatment—as neoplastic cells typically induce such activation both in vitro and in vivo. Our interpretation is that activation likely depends on angiogenic factors: when angiogenesis is blocked pharmacologically or absent due to co-option, the endothelium resembles that of normal lung tissue. The authors suggested, plausibly, that the co-opted endothelium may induce pericyte quiescence and resistance to activation. Finally, they examined immune cells and their interaction with the vascular microenvironment. In Sunitinib-treated mice, macrophages displayed an enhanced matrix-remodelling phenotype, which may in turn facilitate vascular co-option by metastases.

## 3. Co-Option in the Liver: Primary and Metastatic Lesions

Primary advanced hepatocellular carcinomas are rare—at least in the Western world—and highly aggressive. They are potentially treated with the anti-angiogenic drug Sorafenib, though the clinical benefit has been lower than anticipated. Sorafenib inhibits vascular endothelial growth factor receptor 2 (VEGFR2), PDGFR, and Raf kinases [14]. In mice bearing orthotopic liver tumours, resistance emerges after approximately one month, driven by an increase in vascular co-option. Up to 75% of total intra-tumoral vessels were co-opted in treated animals, compared to 23% in untreated controls. Moreover, tumours with higher levels of co-option exhibited increased epithelial-to-mesenchymal transition (EMT), associated with more aggressive behaviour. Notably, when Sorafenib treatment was discontinued, tumours reverted to angiogenesis and reduced invasiveness, including partial regression of EMT [14].

Yang et al. reviewed the hepatic niche to investigate how both primary liver tumours and metastases may promote vascular co-option [15]. They described three immune phenotypes: immune desert, immune-inflamed, and immune-excluded. In the liver, co-option is associated with the immune desert phenotype, characterized by a low density of T cells, which correlates with poor response to therapy. Tumour-associated macrophages assist neoplastic cells in navigating the stroma surrounding blood vessels during co-option. Notably, macrophages expressing the GPNMB (GlycoProtein Nonmetastatic Melanoma protein B) gene, involved in cell adhesion and immune response, may inhibit T-cell function while promoting co-option and thus represent a potential therapeutic target. The authors also proposed that hepatic stellate cells and the extracellular matrix (ECM)—including certain Matrix Metallo Proteinases, collagen isoforms, and L1CAM (L1 Cell Adhesion Molecule), which are expressed more in inflammatory conditions, could be modulated to disrupt co-option. A high density of neutrophils was also noted in co-opted lesions.

Intrahepatic cholangiocarcinoma (iCCA), a primary tumour of the biliary tract, is classified histopathologically into Desmoplastic, Pushing, and Replacing growth patterns [16], which also reflect differences in vascular architecture. The Replacing pattern exhibits some limited areas of angiogenesis—evidenced by positivity for markers of newly formed vessels—as well as areas lacking angiogenesis [16,17]. When vascularity was correlated with iCCA subtypes, the Desmoplastic pattern was predominant in Large Bile Duct iCCA (55%), while the Replacing pattern, i.e., roughly equal areas of angiogenesis and the lack thereof, dominated in Cholangiocarcinoma (82.9%) and in Small Bile Duct iCCA. In total, 76 cases from these subtypes were evaluated [16]. Using a slightly different iCCA classification, Nakanuma et al. [18] reported similar findings and concluded that vessel characterization represents a promising new approach in iCCA research.

In contrast to primary liver cancers, metastatic lesions in the liver are common. Among them, those originating from colorectal cancer are the most prevalent, representing approximately one-quarter of colorectal cancer cases [19]. These metastases are the most extensively studied with respect to vascular patterns, following the same histopathological guidelines [17]. The rationale for evaluating vascular features in liver metastases stems from the relevance of anti-angiogenic therapies, despite their limited clinical success.

In a recent review of colorectal liver metastases, Haas et al. confirmed that non-angiogenic metastases exhibit more aggressive behaviour and worse survival outcomes [19]. They emphasized that micro-vessel density should be considered one of several prognostic indicators rather than a stand-alone marker.

The CIB1 (Calcium and Integrin-Binding protein 1) gene, which links calcium signalling and integrins, is involved in diverse cellular processes such as adhesion, apoptosis, and calcium regulation. CIB1 is implicated in both vascular co-option and poor prognosis. Its expression is elevated in early-stage and right-sided colorectal tumours—the latter typically being more aggressive. When liver metastases were classified by a vascular pattern [17], non-angiogenic metastases displaying the replacement pattern were found to be CIB1-positive and associated with worse outcomes [20].

Single-cell sequencing of colorectal liver metastases revealed WNT signalling to be enriched in the replacement histopathological growth pattern (rHGP), alongside high β-catenin expression. In contrast, the WNT inhibitors DKK1 and DKK4 had an expression elevated in desmoplastic HGP (dHGP), which is consistent with the existing literature [21]. Given the known role of WNT in promoting aerobic glycolysis and the pentose phosphate pathway (PPP), the authors also investigated glycolysis-related pathways and found upregulated proteins in non-angiogenic tumours [21]. Similarly, metabolomic profiling in a model of non-angiogenic lung metastases from renal cancer cells revealed increased glycolysis, although firm conclusions could not be drawn [22].

Breast cancer is another frequent source of liver metastases. Transcriptomic analysis of ten such metastases, encompassing both the replacement and desmoplastic patterns, revealed distinct gene expression profiles. In replacement-pattern metastases, gene clusters involved in cell cycle regulation, extracellular matrix organization, axon guidance, and actin-based processes were enriched. In contrast, desmoplastic regions showed upregulation of genes associated with stress response, immunity, and wound healing [23].

Long non-coding RNAs (lncRNAs) were also implicated in vascular co-option in a study on colorectal cancer liver metastases. One such lncRNA, SYTL5-OT4, was found to promote co-option by preventing the autophagic degradation of ASCT2 (Alanine, Serine, Cysteine Transporter 2) [21]. SYTL5-OT4 was identified as a top differentially expressed lncRNA in Bevacizumab-resistant HCT116 cells and was also upregulated in cells resistant to Regorafenib, a tyrosine kinase inhibitor targeting angiogenesis. Resistance was associated with the induction of vascular co-option. SYTL5-OT4 mediates this effect by stabilizing ASCT2 (Alanine–Serine–Cysteine Transporter 2), a glutamine transporter known to be upregulated in hypoxic cells. ASCT2 supports cell motility, EMT, proliferation, and viability, and is essential for co-option, making it a critical mediator of SYTL5-OT4 function.

The immune response plays a central role in non-angiogenic tumours and is a focal point of the review by Palmieri et al. [24]. In colorectal liver metastases, the replacement pattern typically presents as an immune desert with minimal immune cell infiltration, in stark contrast to the immune-inflamed desmoplastic pattern, which is rich in CD8+ T cells. Interestingly, high mRNA levels of LOXL4, involved in the genesis of connective tissue, were detected in the limited neutrophil population present in these non-angiogenic metastases [19]. LOXL4 has been confirmed as a relevant marker in the replacement type and is believed to facilitate further metastatic spread supporting its potential as a biomarker for this subtype of colorectal liver metastases.

## 4. Co-Option in the Brain: Primary and Metastatic Lesions

The brain is also a well-recognized site for vascular co-option, which has been observed in both primary and metastatic tumours. The authors of a recent study [25] investigated a glioblastoma patient-derived cell line, IDH-wt MGG4, and identified a subpopulation, termed CL3 (based on IKAP classification), which increased following gamma irradiation—likely due to the plasticity of the irradiated cells. These CL3 cells were enriched in Nestin expression. A similar effect was observed with chemotherapy using Temozolomide, a DNA-damaging agent like gamma irradiation, and the combination of both treatments produced a cumulative increase in CL3 cells. Transcriptional profiling revealed that CL3 cells exhibited signatures consistent with slow cycling and senescence-like phenotypes, the latter typically triggered by DNA damage.

To examine the microenvironment in which these tumour cells grow, the authors examined murine brain slices and developed an assay to evaluate co-option. They found that CL3 cells, enriched in Nestin and resistant to treatment, exhibited non-angiogenic growth along pre-existing vessels. Notably, cells initially low in Nestin upregulated this marker when exposed to the perivascular niche. These findings demonstrate that blood vessels, through their angiocrine signalling and conditioned microenvironment, influence tumour cell behaviour and promote co-option. This process is mediated through the FGFR1–YAP1 axis, which is involved in the formation of connective tissue, which also contributes to anti-angiogenic therapy resistance [25].

Uroz et al. [26] provided further insight into non-angiogenic brain metastases. Using in vitro models—including brain slices and a microfluidic device in the absence of the angiogeneic VEGF—they recreated the vascular and perivascular environment characteristic of co-option. The study demonstrated that Talin 1 and Talin 2, proteins involved in adhesion and signalling, regulate tumour cell adhesion during co-option in the brain. Deletion of both Talins significantly impaired tumour cell movement. Furthermore, their microfluidic system revealed that the differential stiffness between the more rigid vasculature and the softer cerebral parenchyma serves as a key driver of co-option. The authors suggested that endothelial cells may facilitate co-option by contributing to vessel stiffness. They concluded by raising an open question: do age-related changes in brain vascular rigidity alter the capacity of neoplastic cells to co-opt and spread within the brain?

## 5. Co-Option in Other Organs

The study of vascular co-option is now expanding beyond traditionally examined organs to include the primary tumours of other sites. One such example is oral squamous-cell carcinoma (OSCC), where co-option has been proposed as a promising therapeutic target [27]. In this retrospective study, vessel co-option was assessed morphologically, based on the presence of strong smooth muscle actin (SMA) staining around the vessels. In contrast, the neoangiogenic vessels exhibited weaker SMA staining. Endothelial cells were marked with CD34, while tumour cells were identified using p40, a subunit of cytokines that contributes to the development of T helper 1 (Th1) cells. Based on these criteria, co-option was demonstrated in 72% of cases (28 out of 39). Proliferative activity, assessed via Ki67, was also analyzed. Tumours with vessel co-option showed higher Ki67 levels (with >10% considered high), and these cases were associated with worse clinical outcomes.

Another organ of interest for studying vascular co-option is the lymph node, especially in the context of lymphomas. Elledge et al. [28] developed an in vitro model using mantle cell lymphoma cell lines co-cultured with human umbilical vein endothelial cells (HUVECs) to generate capillary-like structures (CLSs) in the absence of VEGF. In this system, lymphoma cells aligned with CLSs in a manner suggestive of co-option. Although direct evidence of co-option in human biopsies was not provided, the authors also established an in vivo mouse model in which mantle cell lymphoma infiltrated the lung and also displayed a characteristic vascular co-option pattern.

To therapeutically block this process, the authors engineered a construct termed αCD20-EndoP125A, which links an anti-CD20 antibody to a mutant form of human endostatin containing a proline-to-alanine substitution at position 125. This mutation enhances endostatin’s binding affinity for endothelial cells and increases its anti-angiogenic properties. Not only did EndoP125A inhibit new capillary formation, but the αCD20-EndoP125A construct also impaired lymphoma cell motility, migration, and invasion. Specifically, it disrupted the alignment of lymphoma cells along blood vessels, thereby interfering with co-option. Moreover, the construct reduced the expression of CXCL12 and CXCR4—two key molecules secreted by both neoplastic cells and HUVECs in mono- and co-culture. The CXCL12–CXCR4 axis is critical in both angiogenesis and vascular co-option, suggesting a dual mechanism of therapeutic relevance.

However, the dissection of the contribution of vascular co-option in primary tumours within lymph nodes [29] remains very challenging. This is because such studies are compromised by the extent of angiogenesis in this organ, where the formation of new vessels takes place even under inflammation-induced conditions within “reactive” lymph nodes [30].

## 6. Treatment

Currently, few protocols specifically target vascular co-option. In a relatively small retrospective study of 108 patients with liver metastases from colorectal carcinoma, Rada et al. [31] observed that among 20 patients taking metformin (as an antidiabetic medication), the prevalence of the replacement (non-angiogenic) histopathological growth pattern was lower than in the remaining 88 patients not receiving metformin. The mechanism by which metformin may inhibit co-option remains unclear, but the drug is known to suppress cell proliferation, motility, and the epithelial-to-mesenchymal transition (EMT)—all processes implicated in co-option [31].

In an earlier 2020 study [32], the authors identified co-option as a cause of resistance to therapy in ovarian cancer and aimed to overcome it. They presented results from two clinical trials—a Phase 1b and a Phase 2—testing Pazopanib versus Pazopanib combined with Fosbretabulin. Pazopanib is a tyrosine kinase inhibitor with anti-angiogenic properties, while Fosbretabulin is a vascular disrupting agent that targets microtubules and reduces blood flow. The authors did not assess whether the tumours studied exhibited vessel co-option, although they assumed this to be the case. Therefore, the results should be interpreted with caution.

Co-option is increasingly proposed as a treatment target “in principle”, though direct clinical evidence remains scarce. Ribatti et al. [33] highlighted co-option as a mechanism of resistance to anti-angiogenic therapy and proposed that future strategies combine anti-angiogenic agents with therapies specifically targeting co-option. Similarly, Haas et al. [19] advocated for an integrated approach, suggesting that anti-co-option strategies should complement anti-angiogenic treatments. They emphasized targeting cell motility pathways, particularly the Actin 2/3 complex.

Carrera-Aguado and colleagues [34] advanced this concept by outlining specific therapeutic strategies to target co-option. These include (1) the inhibition of cell adhesion, (2) the modulation of extracellular matrix components, (3) the suppression of cell motility, (4) vascular remodelling, and (5) the inhibition of EMT. They also provided a list of candidate drugs with potential anti-co-option activity.

Kuo et al. [35] discussed several Phase III trials combining VEGF-blocking agents with immune checkpoint inhibitors (ICIs). The rationale behind this combination lies in VEGF’s known immunosuppressive effects—its blockade is thought to enhance ICI efficacy. Although they are non-angiogenic, co-opting tumours are common in both primary and metastatic settings, and it is unclear whether such tumours secrete VEGF at levels sufficient to benefit from these combinations. The question remains largely unexplored. Preliminary studies suggest the combination may not be as effective as expected, particularly in non-angiogenic settings. The authors conclude that further investigation is needed, especially as non-angiogenic tumours remain underrecognized and may constitute a major source of treatment resistance.

Where possible, the pathologist should be asked to recognize such tumours and, in any case, should provide a conclusion on vascularization if anti-vascular drugs are being considered.

## 7. Techniques

To identify co-opted vessels in the lung, which have a characteristic “chicken wire” appearance, Pezzella et al. [5] used CD31. Similarly, Bridgeman et al. [36] used CD31 but combined it with Cytokeratin 7 (CK7), a marker of pneumocytes, and, as a control, Thyroid Transcription Factor 1, which identifies the same cells. Toward the centre of the metastasis, CK7 staining may be lost, indicating the loss of pneumocytes. The CD31-positive capillaries, however, are retained. In the interstitial pattern, which is likewise non-angiogenic, co-staining for CAIX (positive on the renal cancer cells tested) and CK7 for pneumocytes was performed. CD31/CAIX double staining was then carried out to confirm the result.

Teuwen and colleagues performed double staining with anti-CD34 (which, like CD31, identifies all vessels) and, respectively, Endothelial Cell-Specific Molecule 1 (ESM1), which is specific for pulmonary blood capillaries, and Podoplanin, a marker of lymphatics [13] to characterize the lung tumours studied.

Annese et al. [36] investigated vascular co-option in glioblastoma by employing double immunostaining combined with image analysis. Using the Aperio Colocalization algorithm (Leica Biosystems, Buccinasco Italy), they evaluated the expression of P-glycoprotein, a component of the multidrug resistance pump (P-gp)/CD31 and/or S100A10, a protein linked to calcium, /CD31 as markers to study vessel co-option in these tumours. P-gp is expressed on the intact blood–brain barrier in humans and serves as a marker for pre-existing vessels. S100A10, a member of the S100 protein family, is considered a marker of non-stem glioma cells and is implicated in mediating tumour cell–endothelium interactions. Since co-opted endothelial cells exhibit lower proliferation compared to angiogenic endothelium, the authors proposed that regions with low CD31 and P-gp expression—evaluated as mitochondrial localization—surrounded by tumour cells expressing P-gp and/or S100A10 correspond to areas of vascular co-option.

An in vitro three-dimensional tissue-engineered model of breast cancer brain metastasis [37] was developed to study (1) tumour growth regulation, (2) the dynamics of vascular co-option and mosaic vessel formation, (3) the permeability of the blood–tumour barrier (BTB) to therapeutic antibodies, (4) changes in the blood–brain barrier (BBB) and BTB endothelium, and (5) the effects of macrophages co-cultured with BTB endothelial cells. In vivo, vascular co-option was observed in 18% of mouse models. The in vitro model revealed endothelial degeneration as a key feature, with vascular collapse occurring most frequently. Endothelial cells disappeared in these regions, exposing tumour cells directly to the vessel lumen, resulting in mosaic vessel formation. Vascular co-option in this system was observed to occur subsequent to mosaic vessel formation and prior to the final degeneration of the BTB [37].

## 8. Conclusions

Vascular co-option is now widely recognized as an alternative to angiogenesis for tumour growth. Although new blood vessels are not the universal therapeutic target they were presumed to be, Folkman’s foundational contributions are not diminished. His theory—that vessels are important for cancer progression—remains valid. The broader field of cancer vascular biology, rather than angiogenesis alone, continues to recognize him as a pioneer. Our understanding of non-angiogenic tumours has advanced precisely because we stand on the shoulders of this giant.

## Figures and Tables

**Figure 1 cells-14-01490-f001:**
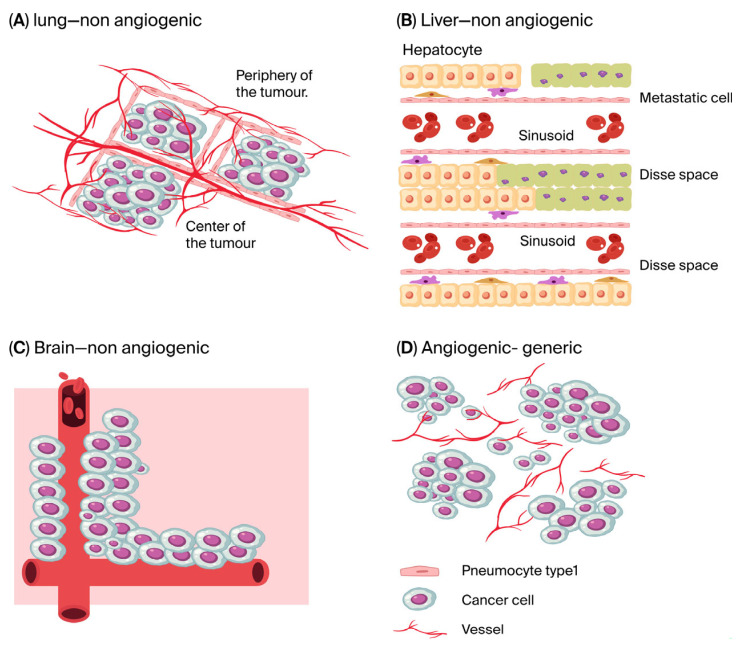
The three main patterns of non-angiogenic tumours, observable both in primary tumours and metastases (**A**–**C**), and the generic scheme for angiogenic ones (**D**), also present in primary and metastatic lesions. (**A**) The main pattern in the lung is called Alveolar, because the neoplastic cells fill the alveoli, taking oxygen from the alveolar vessels. The alveolar walls are covered by type I pneumocytes. Over time, the pneumocytes can be eliminated by the neoplastic cells, which then come into direct contact with the vessels. (**B**) The pattern of co-option, called replacement, is present in the liver. The tumour cells take the place of, or “replace,” the hepatocytes. It is not yet clarified what happens to the displaced hepatocytes. There are two main opinions: one is that they undergo apoptosis, and the other is that they are “phagocytosed” by the advancing tumour cells. (**C**) In the brain, the neoplastic cells arrange themselves along the pre-existing vessels, advancing inside the extracellular neural matrix. In primary tumours, the cells move towards the vessel along a gradient of bradykinin produced by the endothelium. Primary tumour cells reaching the vessel can then “fuse” with the pericytes. Metastatic cells extravasate and are protected from neuroserpin, as this molecule inhibits plasminogen activator, which would otherwise cause apoptosis of the tumour cells via FasL. (**D**) Angiogenic tumours, regardless of the organ in which they originate, cause destruction of the original tissue, including the vessels. Having destroyed the tissue, the overgrowing neoplastic cells become hypoxic and trigger the traditional pathway of angiogenesis, leading to the formation of new vessels.

## Data Availability

No new data were created or analyzed in this study. Data sharing is not applicable to this article.

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
