# Peer review of "A Change of Hallmark: An Update"

_cells, 2025, doi:10.3390/cells14191490_

Round 1
Reviewer 1 Report
Comments and Suggestions for Authors
This review article by Donnem et al. provides a brief update on the most important advances in the phenomenon of vessel co-option in tumors. This is a very nice and useful update of vessel co-option. The authors of the manuscript are leading experts in the field. The review begins by comparing primary and metastatic tumors in the three hypervascularized organs where vessel co-option occurs: the lung, liver, and brain. I find their contribution regarding other organs very interesting, as this is a much less studied area than the main organs.
I have several suggestions for improving some sections:
-In the section on techniques, you could add how VCO growth patterns are examined vs. angiogenesis. For example, in the lung, through the use of epithelial markers such as Podoplanin (as done by Teuwen et al., 2021), or CK7 (as done by Bridgeman et al., 2017).
-On the other hand, add how vessel co-option could be studied in vitro or ex vivo (experiments in the absence of VEGF).
-I would encourage the authors to improve the figure. I think it could provide much more information if they detail other types of growth patterns, as they have done in other high-quality articles that they have published in the last years.
Comments on the Quality of English LanguageThe quality of english language is perfect.
Author Response
1. In the section on techniques, you could add how VCO growth patterns are examined vs. angiogenesis. For example, in the lung, through the use of epithelial markers such as Podoplanin (as done by Teuwen et al., 2021), or CK7 (as done by Bridgeman et al., 2017).
We added, in the section on techniques:
“To identify the co-opted vessels in the lung, with their “chicken wire” appearance, Pezzella et al {Pezzella, 1996 #33} used CD31. Always in the lung, Bridgeman et al.{Bridgeman, 2017 #54} used also CD31 but in combination with Cytokeratin 7 (CK7), a maker of pneumocytes, and, as control, with Thyroid Transcription Fatcor 1, which is picking up also the same cells. Toward the centre of the metastasis the staining for CK7 can be lost indicating that the pneumocytes are lost. The CD31 capillares are instead retained. In the interstitial pattern, always non angiogenic, co staining for CAIX (positive on the renal cancer cells tested) and CK7 for pneumocytes was done. Than also CD31/CAIX double staining was performed do confirm the result.
Doble staining with anti CD34 (like CD31 identify all vessels) and, respectively, Endothelial cell-Specific Molecule 1 (ESM1)
specific for blood capillaries of the lung, and Plodoplanin, against lymphatics, was done by Teuwen and colleagues {Teuwen, 2021 #42} to characterize the lung tumours studied.”
2. On the other hand, add how vessel co-option could be studied in vitro or ex vivo (experiments in the absence of VEGF)
Both of the “in vitro” system cited {Uroz, 2024 #44}{Elledge, 2024 #9} are without VEGF. This has now been specified in the text.
3. I would encourage the authors to improve the figure. I think it could provide much more information if they detail other types of growth patterns, as they have done in other high-quality articles that they have published in the last years.
The figure will be submitted to MDPI’s image service.
Reviewer 2 Report
Comments and Suggestions for Authors
Suggestions for Authors.
In the Review "The Change of Hallmark: An Update," by Donnem et al., the authors illustrate how concepts related to tumor vascularization have evolved over time. They present a model of tumor growth that is independent of the development of new vessels, or neoangiogenesis. This model illustrates vascular co-optation, which is a mechanism by which tumor cells adhere to and replace (at least initially) healthy cells in the tissue parenchyma. Vascular co-optation also explains why anti-angiogenic therapy has unfortunately failed in the treatment of tumors.
The review is well written, although perhaps overly schematic, which makes it difficult for non-experts to understand. It would be helpful if the authors clarified the role of each gene or protein mentioned in the text, briefly explaining how they function under normal conditions or in tumor growth (e.g., p40 IKB, etc… How many readers are familiar with the effects of p40, depending on the chain with which it heterodimerizes or the receptor to which it binds?)
Sezione Treatment
This is probably the weakest section of the manuscript. It appears to be merely a compilation of conclusions from other reviewers without any critical commentary.
Line 268 to 274: the following statement:
“In an earlier 2020 study (31), the authors identified co-option as a cause of resistance to therapy in ovarian cancer and aimed to overcome it. They presented results from two clinical trials—a Phase 1b and a Phase 2—testing Pazopanib versus Pazopanib combined with Fosbretabulin. Pazopanib is a tyrosine kinase inhibitor with anti-angiogenic properties, while Fosbretabulin is a vascular-disrupting agent that targets microtubules and reduces blood flow. However, the authors did not assess whether the tumours studied exhibited vessel co-option”. This citation contains an internal contradiction. If it does not demonstrate that the tumors under examination presented vascular co-option, why did the Authors choose to cite it?
Lines 294-295: the statement: “The authors conclude that further investigation is needed, especially as non-angiogenic tumours remain underrecognized and may constitute a major source of treatment resistance”. Did the authors of this review not ask themselves this question? Could vascular co-optation, particularly in metastases, be a result of anti-angiogenic therapy?
Every tumor, even though it initially grows through vascular co-optation, becomes capable of inducing the growth of new vessels such as neoangiogenesis and adult vasculogenesis. The authors should also cite studies that have addressed this aspect.
Minor point
To enhance the reader's comprehension, the following acronyms should be written in full when they appear in the text:
Line 103: GPNMB
Line 106: LICAM1
Line 131: CIB1
Line 155: ASCT2
Line 167: LOXL4
Line 176: IKAP. Furthermore, explain that this algorithm identifies main cell groups and improves their differentiation through the systematic adjustment of clustering parameters.
Line 174: Ref. 24s should be changed as 24
Author Response
1. The review is well written, although perhaps overly schematic, which makes it difficult for non-experts to understand. It would be helpful if the authors clarified the role of each gene or protein mentioned in the text, briefly explaining how they function under normal conditions or in tumor growth (e.g., p40 IKB, etc… How many readers are familiar with the effects of p40, depending on the chain with which it heterodimerizes or the receptor to which it binds?)
Done.
2. Line 268 to 274: the following statement:
“In an earlier 2020 study (31), the authors identified co-option as a cause of resistance to therapy in ovarian cancer and aimed to overcome it. They presented results from two clinical trials—a Phase 1b and a Phase 2—testing Pazopanib versus Pazopanib combined with Fosbretabulin. Pazopanib is a tyrosine kinase inhibitor with anti-angiogenic properties, while Fosbretabulin is a vascular-disrupting agent that targets microtubules and reduces blood flow. However, the authors did not assess whether the tumours studied exhibited vessel co-option”. This citation contains an internal contradiction. If it does not demonstrate that the tumors under examination presented vascular co-option, why did the Authors choose to cite it?
A valid point, and we have rephrased the statement: “However, the authors did not assess whether the tumours studied exhibited vessel co-option, although they assume this to be the case. Therefore, the results should be interpreted with caution.”
3. Lines 294-295: the statement: “The authors conclude that further investigation is needed, especially as non-angiogenic tumours remain underrecognized and may constitute a major source of treatment resistance”. Did the authors of this review not ask themselves this question? Could vascular co-optation, particularly in metastases, be a result of anti-angiogenic therapy?
The authors describe cohorts in which co-option precedes therapy but also cite therapy-induced co-opting tumours. When they state that further work is needed, they are mainly referring to the fact that non-angiogenic tumours are often not recognized at diagnosis.
We have added the following: ‘Where possible, the pathologist should be asked to recognize such tumours and, in any case, should provide a conclusion on vascularization if anti-vascular drugs are being considered.”
4. Every tumor, even though it initially grows through vascular co-optation, becomes capable of inducing the growth of new vessels such as neoangiogenesis and adult vasculogenesis. The authors should also cite studies that have addressed this aspect.
We have added:
‘Furthermore, it should not be forgotten that several angiogenic tumours begin with co-option of pre-existing vessels rather than as avascular masses. This concept was introduced by Holash and colleagues in 1999 {Holash, 1999 #53}. Subsequently, the co-opted vessels regress, the neoplastic mass becomes avascular, and only then is neo-angiogenesis triggered.”
5. Minor point
To enhance the reader's comprehension, the following acronyms should be written in full when they appear in the text:
Line 103: GPNMB
Line 106: LICAM1
Line 131: CIB1
Line 155: ASCT2
We have added this in full. We apologize for the earlier error: LICAM1 does not exist; the correct name is L1CAM